# A Study of Generalized Laguerre Poly-Genocchi Polynomials

**Nabiullah Khan** [1] , **Talha Usman** [1] **and Kottakkaran Sooppy Nisar** [2],*

[1] Department of Applied Mathematics, Faculty of Engineering and Technology, Aligarh Muslim University, Aligarh 202002, India; nukhanmath@gmail.com (N.K.); talhausman.maths@gmail.com (T.U.)
[2] Department of Mathematics, College of Arts and Science at Wadi Al-dawaser, Prince Sattam Bin Abdulaziz University, Wadi Al-dawaser 11991, Saudi Arabia
* Correspondence: n.sooppy@psau.edu.sa; Tel.: +966-563456976

**Abstract:** A variety of polynomials, their extensions, and variants, have been extensively investigated, mainly due to their potential applications in diverse research areas. Motivated by their importance and potential for applications in a variety of research fields, numerous polynomials and their extensions have recently been introduced and investigated. In this paper, we introduce generalized Laguerre poly-Genocchi polynomials and investigate some of their properties and identities, which were found to extend some known results. Among them, an implicit summation formula and addition-symmetry identities for generalized Laguerre poly-Genocchi polynomials are derived. The results presented here, being very general, are pointed out to reduce to yield formulas and identities for relatively simple polynomials and numbers.

**Keywords:** Laguerre polynomials; hermite polynomials; Laguerre poly-Genocchi polynomials; summation formulae; symmetric identities

## 1. Introduction

The generating function of the two variable Laguerre polynomials $L_p(u,v)$ [1] is defined by

$$e^{v\zeta}C_0(u\zeta) = \sum_{p=0}^{\infty} L_p(u,v)\frac{\zeta^p}{p!}, \tag{1}$$

where $C_0(u)$ is the 0-th order Tricomi function [2]

$$C_0(u) = \sum_{s=0}^{\infty} \frac{(-1)^s u^s}{(s!)^2} \tag{2}$$

we find from Equation (1) that

$$L_p(u,v) = \sum_{r=0}^{p} \frac{p!(-1)^r v^{p-r}u^r}{(p-r)!(r!)^2} \tag{3}$$

The generalized Bernoulli $B_p^\alpha(u)$, Euler $E_p^\alpha(u)$ and Genocchi $G_p^\alpha(u)$ polynomials of order $\alpha$, each of degree $p$ are defined, respectively, by the following generating functions (see References [1–30]):

$$\left(\frac{\zeta}{e^\zeta - 1}\right)^\alpha e^{u\zeta} = \sum_{p=0}^\infty B_p^\alpha(u)\frac{\zeta^p}{p!} \qquad (|\zeta| < 2\pi; 1^\alpha = 1) \tag{4}$$

$$\left(\frac{2}{e^\zeta + 1}\right)^\alpha e^{u\zeta} = \sum_{p=0}^\infty E_p^\alpha(u)\frac{\zeta^p}{p!} \qquad (|\zeta| < \pi; 1^\alpha = 1) \tag{5}$$

$$\left(\frac{2\zeta}{e^\zeta + 1}\right)^\alpha e^{u\zeta} = \sum_{p=0}^\infty G_p^\alpha(u)\frac{\zeta^p}{p!} \qquad (|\zeta| < \pi; 1^\alpha = 1) \tag{6}$$

It is easy to see that

$$B_p(u) = B_p^{(1)}(u) = B_p, \ E_p(u) = E_p^{(1)}(u), \ G_p(u) = G_p^{(1)}(u) \tag{7}$$

for Bernoulli number $B_p$, Euler number $E_p$, and Genocchi number $G_p$,

$$B_p^{(1)}(0) = B_p(0) = B_p, \ E_p^{(1)}(0) = E_p(0) = E_p, \ G_p^{(1)}(0) = G_p(0) = G_p. \tag{8}$$

Recently, Kaneko [3] introduced and investigated generalized poly-Bernoulli numbers by means of the generating functions

$$\frac{Li_k(1 - e^{-\zeta})}{1 - e^{-\zeta}} = \sum_{p=0}^\infty B_p^{(k)}\frac{\zeta^p}{p!} \tag{9}$$

where $k \in w$ and

$$Li_w = \sum_{q=1}^\infty \frac{w^q}{q!}, \ |w| < 1$$

so for $k \le 1$,

$$Li_k = -\ln(1 - w), \ Li_0(w) = \frac{w}{1 - w}, \ Li_{-1} = \frac{w}{(1 - w)^2}, \ \cdots$$

When $k \le 1$, the left-hand side of (1.9) can be expressed as

$$\underbrace{e^\zeta \frac{1}{e^\zeta - 1}\int_0^\zeta \frac{1}{e^\zeta - 1} \cdots \int_0^\zeta \frac{1}{e^\zeta - 1}\int_0^\zeta \frac{1}{e^\zeta - 1}} \ d\zeta d\zeta \cdots d\zeta = \sum_{p=0}^\infty B_p^{(k)}\frac{\zeta^p}{p!}$$

in the special case, one can see

$$B_p^{(1)} = B_p.$$

in References [4–7], Jolany et al. introduced and studied the generalized poly-Bernoulli numbers and polynomials, which appear in the following power series

$$\frac{Li_k(1 - (\alpha\beta)^{-\zeta})}{\beta^\zeta - \alpha^{-\zeta}} = \sum_{p=0}^\infty B_p^{(k)}(\alpha, \beta)\frac{\zeta^p}{p!}, \ |\zeta| < \frac{2\pi}{|\ln ff + \ln fi|} \tag{10}$$

$$\frac{Li_k(1 - (\alpha\beta)^{-\zeta})}{\beta^\zeta - \alpha^{-\zeta}}e^{u\zeta} = \sum_{p=0}^\infty B_p^{(k)}(u, \alpha, \beta)\frac{\zeta^p}{p!}, \ |\zeta| < \frac{2\pi}{|\ln ff + \ln fi|} \tag{11}$$

$$\frac{Li_k(1 - (\alpha\beta)^{-\zeta})}{\beta^\zeta - \alpha^{-\zeta}}c^{u\zeta} = \sum_{p=0}^\infty B_p^{(k)}(u, \alpha, \beta, c)\frac{\zeta^p}{p!}, \ |\zeta| < \frac{2\pi}{|\ln ff + \ln fi|} \tag{12}$$

one can easily see that

$$B_p^{(k)}(0, 1, e) = B_p^{(k)}, \ B_p^{(k)}(u) = 1 + u$$

and

$$B_p^{(k)}(u) = B_p^{(k)}(e^{u+1}, e^u) \tag{13}$$

where $B_p^{(k)}$ are generalized poly-Bernoulli numbers. For synchronized works about poly-Bernoulli polynomials, see References [3–6].

The two-variable Hermite Kampé de Fériet generalization of the Hermite polynomials (see References [8,9,11])

$$H_p(u,v) = p! \sum_{s=0}^{[\frac{p}{2}]} \frac{v^s u^{p-2s}}{s!(p-2s)!} \tag{14}$$

they are usually defined by the following generating function

$$e^{u\zeta + v\zeta^2} = \sum_{p=0}^{\infty} H_p(u,v) \frac{\zeta^p}{p!} \tag{15}$$

polynomials $H_p(u,v)$ in Equation (15) reduce to ordinary Hermite polynomials $H_p(u)$ when $v = -1$ and $u$ is replaced by $2u$.

Pathan et al. [10] introduced and investigated the generating function for the generalized Hermite–Bernoulli polynomials of two variables ${}_H B_p^{(\alpha)}(u,v)$ as follows

$$\left(\frac{\zeta}{e^\zeta - 1}\right)^\alpha e^{u\zeta + v\zeta^2} = \sum_{p=0}^{\infty} {}_H B_p^{(\alpha)}(u,v) \frac{\zeta^p}{p!} \tag{16}$$

case $\alpha = 1$ in Equation (16) reduces to Hermite–Bernoulli polynomials ${}_H B_p(u,v)$ (see Reference [11])

$$\left(\frac{\zeta}{e^\zeta - 1}\right) e^{u\zeta + v\zeta^2} = \sum_{p=0}^{\infty} {}_H B_p(u,v) \frac{\zeta^p}{p!} \tag{17}$$

the stirling number of the first kind is given by

$$(u)_p = u(u-1)\cdots(u-p+1) = \sum_{r=0}^{p} S_r(p,r)u^r, \quad (n \geq 0) \tag{18}$$

and the stirling number of the second kind is defined as follows

$$(e^\zeta - 1)^p = p! \sum_{r=p}^{\infty} S_2(r,p) \frac{\zeta^r}{r!} \tag{19}$$

A variety of polynomials, their extensions, and variants have been extensively investigated mainly due to their potential applications in diverse research areas. Motivated by their importance and potential for applications in certain problems in number theory, combinatorics, classical and numerical analysis and other fields of applied mathematics, a number of certain numbers and polynomials, and their generalizations have recently been extensively investigated.

The organization of this paper is given as follows:

In Section 2, we introduce generalized Laguerre poly-Genocchi polynomials ${}_L G_p^{(k)}(u,v,w)$ and develop elementary properties by using generating functions for the numbers. In Section 3, we derive implicit summation formulae for these generalized polynomials by using different analytical means on their respective generating functions. In Section 4, we establish addition-symmetry identities by using different analytical means and using generating functions.

## 2. Generalized Laguerre Poly-Genocchi Polynomials $_LG_p^{(k)}(u,v,w)$

In this section, we introduce and investigate the generalized Laguerre poly-Genocchi polynomials as follows:

$$\frac{2Li_k(1-e^{-\zeta})}{e^\zeta+1}\,e^{v\zeta+w\zeta^2}C_0(u\zeta) = \sum_{p=0}^{\infty}{}_LG_p^{(k)}(u,v,w)\frac{\zeta^p}{p!},\quad (k\in w) \tag{20}$$

so that

$$_LG_p^{(k)}(u,v,w) = \sum_{q=0}^{p}\sum_{k=0}^{[\frac{p}{2}]}\frac{G_{p-q}^{(k)}\,L_{q-2k}(u,v)w^k p!}{(q-2k)!k!(p-q)!} \tag{21}$$

when $u=v=w=0$, $G_p^{(k)}=G(0,0,0)$ are known as poly-Genocchi numbers. Using Equation (20), we easily obtain $G_p^{(k)}=0$. Setting $k=1$ in Equation (21), we obtain

$$\frac{2Li_1(1-e^{-\zeta})}{e^\zeta+1}e^{v\zeta+w\zeta^2}C_0(u\zeta) = \sum_{p=0}^{\infty}{}_LG_p(u,v,w)\frac{\zeta^p}{p!},\quad (k\in w) \tag{22}$$

thus, with Equations (20) and (22), we get

$$_LG_p^{(k)}(u,v,w) = {}_LG_p(u,v,w),\quad (p\geq 0).$$

Setting $u=0$ in (20), we obtain the Hermite poly-Genocchi polynomials by Khan [12], defined as

$$\frac{2Li_k(1-e^{-\zeta})}{e^\zeta+1}e^{v\zeta+w\zeta^2} = \sum_{p=0}^{\infty}{}_HG_p^{(k)}(v,w)\frac{\zeta^p}{p!},\quad (k\in w) \tag{23}$$

**Theorem 1.** *For $p\geq 0$, we have*

$$_LG_p^{(2)}(u,v,w) = \sum_{q=0}^{p}\binom{p}{q}\frac{B_q}{q+1}\,{}_LG_{p-q}(u,v,w) \tag{24}$$

**Proof.** Using Definition (20), we have

$$\frac{2Li_k(1-e^{-\zeta})}{e^\zeta+1}\,e^{v\zeta+w\zeta^2}C_0(u\zeta) = \sum_{p=0}^{\infty}{}_LG_p^{(k)}(u,v,w)\frac{\zeta^p}{p!}$$

$$= \left(\frac{2}{e^\zeta+1}\right)\,e^{v\zeta+w\zeta^2}C_0(u\zeta)\underbrace{\int_0^\zeta\frac{1}{e^w-1}\int_0^\zeta\frac{1}{e^w-1}\cdots\frac{1}{e^w-1}\int_0^\zeta\frac{w}{e^w-1}\,dwdw\cdots dw}$$

in particular, $k=2$, we have

$$_LG_p^{(2)}(u,v,w) = \left(\frac{2}{e^\zeta+1}\right)\,e^{v\zeta+w\zeta^2}C_0(u\zeta)\int_0^\zeta\frac{w}{e^w-1}$$

$$= \left(\sum_{q=0}^{\infty}\frac{B_q\zeta^q}{q+1}\right)\left(\frac{2\zeta}{e^\zeta+1}\right)\,e^{v\zeta+w\zeta^2}C_0(u\zeta)$$

$$= \left(\sum_{q=0}^{\infty}\frac{B_q\zeta^q}{q+1}\right)\left(\sum_{p=0}^{\infty}{}_LG_p(u,v,w)\frac{\zeta^p}{p!}\right)$$

replacing $p$ by $p - q$ in the above equation, we have

$$= \sum_{p=0}^{\infty} \sum_{q=0}^{p} \binom{p}{q} \frac{B_q}{q+1} \, {}_L G_{p-q}(u,v,w) \frac{\zeta^p}{p!}$$

On comparing the coefficients of $\zeta$ in the above equation, we obtain Equation (24). □

**Theorem 2.** *For $p \geq 1$, we have*

$$\,_L G_p^{(k)}(u,v,w) = \sum_{q=0}^{p} \sum_{k=0}^{[\frac{p}{2}]} \frac{G_q^{(k)} L_{p-q-2k}(u,v) w^k p!}{q! k! (p-q-2k)!} \tag{25}$$

**Proof.** Using Equation (2.1), we obtain

$$\sum_{p=0}^{\infty} {}_L G_p^{(k)}(u,v,w) \frac{\zeta^p}{p!} = \frac{2 Li_k(1 - e^{-\zeta})}{e^{\zeta}+1} \, e^{v\zeta + w\zeta^2} C_0(u\zeta)$$

$$= \left( \sum_{q=0}^{\infty} G_q^{(k)} \frac{\zeta^q}{q!} \right) \left( \sum_{p=0}^{\infty} \sum_{k=0}^{[\frac{p}{2}]} \frac{L_{p-2k}(u,v) w^k}{k!(p-2k)!} \zeta^p \right)$$

Replacing $p$ with $p - q$ in the above equation, and equating the coefficients of $\zeta^p$, we obtain Equation (25). □

**Theorem 3.** *For $p \geq 0$, we have*

$$\,_L G_p^{(k)}(u,v,w) = \sum_{n=0}^{p} \sum_{r=1}^{n+1} \frac{(-1)^{r+n+1} \, r! S_2(n+1,r)}{r^k (n+1)} \binom{p}{n} \, {}_L G_{p-n}(u,v,w) \tag{26}$$

**Proof.** From Equation (20), we have

$$\sum_{p=0}^{\infty} {}_L G_p^{(k)}(u,v,w) \frac{\zeta^p}{p!} = \left( \frac{Li_k(1 - e^{-\zeta})}{\zeta} \right) \left\{ \left( \frac{2\zeta}{e^{\zeta}+1} \right) e^{v\zeta + w\zeta^2} C_0(u\zeta) \right\} \tag{27}$$

Now

$$\begin{aligned}
\frac{1}{\zeta} Li_k(1 - e^{-\zeta}) &= \frac{1}{\zeta} \sum_{r=1}^{\infty} \frac{(1 - e^{-\zeta})^r}{r^k} = \frac{1}{\zeta} \sum_{r=1}^{\infty} \frac{(-1)^r}{r^k} (1 - e^{-\zeta})^r \\[2mm]
&= \frac{1}{\zeta} \sum_{r=1}^{\infty} \frac{(-1)^r}{r^k} r! \sum_{n=r}^{\infty} (-1)^n S_2(n,r) \frac{\zeta^n}{n!} \\[2mm]
&= \frac{1}{\zeta} \sum_{n=1}^{\infty} \sum_{r=1}^{n} \frac{(-1)^{r+n}}{r^k} l! S_2(n,r) \frac{\zeta^n}{n!} \\[2mm]
&= \sum_{n=0}^{\infty} \left\{ \sum_{r=1}^{n+1} \frac{(-1)^{r+n+1}}{r^k} l! \frac{S_2(n+1,r)}{n+1} \right\} \frac{\zeta^n}{n!}
\end{aligned} \tag{28}$$

From Equations (27) and (28), we get

$$\sum_{p=0}^{\infty} {}_L G_p^{(k)}(u,v,w) \frac{\zeta^p}{p!} = \sum_{n=0}^{\infty} \left\{ \sum_{r=1}^{n+1} \frac{(-1)^{r+n+1}}{r^k} r! \frac{S_2(n+1,r)}{n+1} \right\} \frac{\zeta^n}{n!} \left( \sum_{p=0}^{\infty} {}_L G_p(u,v,w) \frac{\zeta^p}{p!} \right)$$

Replacing $p$ by $p - n$ in r.h.s of above equation and equating the coefficients of $\zeta^p$, we obtain (26). $\square$

**Theorem 4.** *For $p \geq 1$, we have*

$$
{}_L G_p^{(k)}(u, v + 1, w) + {}_L G_p^{(k)}(u, v, w)
$$

$$
= 2 \sum_{n=1}^{p} \sum_{r=1}^{n} \sum_{k=0}^{[\frac{p}{2}]} \frac{(-1)^{r+n}}{r^k} r! p! S_2(n, l) \frac{L_{p-n-2k}(u,v) w^k}{n! k! (p-n-2k)!}
$$

(29)

**Proof.** Applying Definition (2.1), we obtain

$$
\sum_{p=0}^{\infty} {}_L G_p^{(k)}(u, v + 1, w) \frac{\zeta^p}{p!} + \sum_{p=0}^{\infty} {}_L G_p^{(k)}(u, v, w) \frac{\zeta^p}{p!}
$$

$$
= \frac{2 Li_k(1 - e^{-\zeta})}{e^{\zeta} + 1} e^{(v+1)\zeta + w\zeta^2} C_0(u\zeta) + \frac{2 Li_k(1 - e^{-\zeta})}{e^{\zeta} + 1} e^{v\zeta + w\zeta^2} C_0(u\zeta)
$$

$$
= 2 Li_k(1 - e^{-\zeta}) e^{v\zeta + w\zeta^2} C_0(u\zeta)
$$

$$
= \sum_{n=1}^{\infty} \left\{ 2 \sum_{r=1}^{n} \frac{(-1)^{r+n}}{r^k} r! S_2(n, r) \right\} \frac{\zeta^n}{n!} e^{v\zeta + w\zeta^2} C_0(u\zeta)
$$

$$
= \sum_{n=1}^{\infty} \left\{ 2 \sum_{r=1}^{n} \frac{(-1)^{r+n}}{r^k} r! S_2(n, r) \right\} \frac{\zeta^n}{n!} \left( \sum_{p=0}^{\infty} \sum_{k=0}^{[\frac{p}{2}]} \frac{L_{p-2k}(u,v) w^k}{k! (p - 2k)!} \zeta^p \right)
$$

Replacing $p$ by $p - n$ in the above equation and equating the coefficients of $\zeta^p$, we obtain Equation (29). $\square$

**Theorem 5.** *For $d \in N$ with $d \equiv 1 (mod 2)$, we have*

$$
{}_L G_p^{(k)}(u, v, w) = \sum_{n=0}^{p} \binom{p}{n} d^{p-n-1} \sum_{r=0}^{n+1} \sum_{a=0}^{d-1} \frac{(-1)^{r+n+1} r! S_2(n+1, r)}{r^k} (-1)^a {}_L G_{p-n} \left( u, \frac{a+v}{d}, w \right) \quad (30)
$$

**Proof.** From Equation (20), we have

$$
\sum_{p=0}^{\infty} {}_L G_p^{(k)}(u, v, w) \frac{\zeta^p}{p!} = \frac{2 Li_k(1 - e^{-\zeta})}{e^{\zeta} + 1} e^{v\zeta + w\zeta^2} C_0(u\zeta)
$$

$$
= \left( \frac{2 Li_k(1 - e^{-\zeta})}{\zeta} \right) \left\{ \frac{2\zeta}{e^{d\zeta} + 1} \sum_{a=0}^{d-1} e^{(a+v)\zeta + w\zeta^2} C_0(u\zeta) \right\}
$$

$$
= \left\{ \sum_{n=0}^{\infty} \left( \sum_{r=1}^{n+1} \frac{(-1)^{r+n+1}}{r^k} r! \frac{S_2(n+1, r)}{n+1} \right) \frac{\zeta^n}{n!} \right\} \left\{ \sum_{q=0}^{\infty} d^{q-1} \sum_{a=0}^{d-1} (-1)^a {}_L G_p \left( u, \frac{a+v}{d}, w \right) \frac{\zeta^p}{p!} \right\}
$$

Replacing $p$ with $p - n$ in the above equation and equating the coefficient of $\zeta^p$, we obtain Equation (30). $\square$

## 3. Implicit Summation Formulae

Here, we present an interesting implicit summation formulae for generalized Laguerre poly-Genocchi polynomials, ${}_L G_p^{(k)}(u, v, w)$ in Equation (20).

**Theorem 6.** *For $u, v, w \in \Re$ and $p \geq 0$, Then*

$$
{}_LG_{r+n}^{(k)}(u, y, w) = \sum_{p,q=0}^{r,n} \binom{r}{q} \binom{n}{p} (y-v)^{p+q} {}_LG_{r+n-p-q}^{(k)}(u, v, w) \tag{31}
$$

**Proof.** Replacing $\zeta$ with $\zeta + x$ in Equation (20), we have

$$
\left\{ \frac{2Li_k(1 - e^{-(\zeta+x)})}{e^{\zeta+x} + 1} \right\} e^{w(\zeta+x)^2} C_0(u(\zeta+x)) = e^{-v(\zeta+x)} \sum_{r,n=0}^{\infty} {}_LG_{r+n}^{(k)}(u, v, w) \frac{\zeta^r}{r!} \frac{x^n}{n!} \tag{32}
$$

We find that the left member of Equation (32) is independent of variable $v$. In this regard, replacing $v$ by another variable $y$ in the right member of Equation (32) and equating the right members of the resulting identity and Equation (32), we get

$$
e^{(y-v)(\zeta+x)} \sum_{r,n=0}^{\infty} {}_LG_{r+n}^{(k)}(u, v, w) \frac{\zeta^r}{r!} \frac{x^n}{n!} = \sum_{r,n=0}^{\infty} {}_LG_{r+n}^{(k)}(u, y, w) \frac{\zeta^r}{r!} \frac{x^n}{n!} \tag{33}
$$

elaborating exponential function in Equation (33), we obtain

$$
\sum_{P=0}^{\infty} \frac{[(y-v)(\zeta+x)]^P}{P!} \sum_{r,n=0}^{\infty} {}_LG_{r+n}^{(k)}(u, v, w) \frac{\zeta^r}{r!} \frac{x^n}{n!} = \sum_{r,n=0}^{\infty} {}_LG_{r+n}^{(k)}(u, y, w) \frac{\zeta^r}{r!} \frac{x^n}{n!} \tag{34}
$$

now, by using expression (Reference [13], p. 52(2))

$$
\sum_{P=0}^{\infty} f(P) \frac{(u+v)^P}{P!} = \sum_{p,q=0}^{\infty} f(p+q) \frac{u^p}{p!} \frac{v^q}{q.!} \tag{35}
$$

the left-hand side of Equation (35) becomes

$$
\sum_{p,q=0}^{\infty} \frac{(y-v)^{p+q} \zeta^q x^p}{p!q!} \sum_{r,n=0}^{\infty} {}_LG_{r+n}^{(k)}(u, v, w) \frac{\zeta^r}{r!} \frac{x^n}{n!} = \sum_{r,n=0}^{\infty} {}_LG_{r+n}^{(k)}(u, y, w) \frac{\zeta^r}{r!} \frac{x^n}{n!} \tag{36}
$$

replacing $r$ by $r - q$, $n$ by $n - p$ and using the lemma (Reference [13], p. 100(1)) in Equation (36), we obtain

$$
\sum_{p,q=0}^{\infty} \sum_{r,n=0}^{\infty} \frac{(y-v)^{p+q}}{p!q!} {}_LG_{r+n-p-q}^{(k)}(u, v, w) \frac{\zeta^r}{(r-q)!} \frac{x^n}{(n-p)!} = {}_LG_{r+n}^{(k)}(u, y, w) \frac{\zeta^r}{r!} \frac{x^n}{n!} \tag{37}
$$

now, equating the coefficients of $\zeta$ and $x$ on both sides of Equation (37), we obtain Equation (31). $\square$

**Theorem 7.** *For $u, v, w \in \Re$ and $p \geq 0$. Then,*

$$
{}_LG_p^{(k)}(u, v+x, w) = \sum_{j=0}^{p} \binom{p}{j} x^j {}_LG_{p-j}^{(k)}(u, v, w) \tag{38}
$$

**Proof.** Since

$$
\sum_{p=0}^{\infty} {}_LG_p^{(k)}(u, v+x, w) \frac{\zeta^p}{p!} = \frac{2Li_k(1 - e^{-\zeta})}{e^\zeta + 1} e^{(v+x)\zeta + w\zeta^2} C_0(u\zeta)
$$

$$
\sum_{p=0}^{\infty} {}_LG_p^{(k)}(u, v+x, w) \frac{\zeta^p}{p!} = \left( \sum_{p=0}^{\infty} {}_LG_p^{(k)}(u, v, w) \frac{\zeta^p}{p!} \right) \left( \sum_{j=0}^{\infty} x^j \frac{\zeta^j}{j!} \right)
$$

replacing $p$ with $p - j$ and equating the coefficients of $\zeta^p$, we obtain Equation (38). $\square$

**Theorem 8.** *For $u, v, w \in \Re$ and $p \geq 0$. Then,*

$$_L G_p^{(k)}(u, v + x, w + z) = \sum_{q=0}^{p} \binom{p}{q} {}_L G_{p-q}^{(k)}(u, v, w) H_q(x, z) \tag{39}$$

**Proof.** By the definition of generalized Laguerre poly-Genocchi polynomials and Definition (15), we have

$$\frac{2 Li_k(1 - e^{-\zeta})}{e^\zeta + 1} e^{(v+x)\zeta + (w+z)\zeta^2} C_0(u\zeta) = \left( \sum_{p=0}^{\infty} {}_L G_p^{(k)}(u, v, w) \frac{\zeta^p}{p!} \right) \left( \sum_{q=0}^{\infty} H_q(x, z) \frac{\zeta^q}{q!} \right)$$

replacing $p$ with $p - q$ and equating the coefficients of $\zeta^p$, we obtain Equation (39).  □

**Theorem 9.** *For $u, v, w \in \Re$ and $p \geq 0$. Then,*

$$_L G_p^{(k)}(u, v, w) = \sum_{q=0}^{p-2j} \sum_{j=0}^{[\frac{p}{2}]} \frac{G_q^{(k)} L_{p-q-2j}(u, v) w^j p!}{q! j! (p - q - 2j)!} \tag{40}$$

**Proof.** Applying Definition (20) to the term $\frac{2 Li_k(1-e^{-\zeta})}{e^\zeta + 1}$ and expanding exponential and tricomi function $e^{y\zeta + z\zeta^2} C_0(x\zeta)$ at $\zeta = 0$ yields

$$\frac{2 Li_k(1 - e^{-\zeta})}{e^\zeta + 1} e^{v\zeta + w\zeta^2} C_0(u\zeta) = \left( \sum_{q=0}^{\infty} G_q^{(k)} \frac{\zeta^q}{q!} \right) \left( \sum_{p=0}^{\infty} L_p(u, v) \frac{\zeta^p}{p!} \right) \left( \sum_{j=0}^{\infty} w^j \frac{\zeta^{2j}}{j!} \right)$$

$$\sum_{p=0}^{\infty} {}_L G_p^{(k)}(u, v, w) \frac{\zeta^p}{p!} = \sum_{p=0}^{\infty} \left( \sum_{q=0}^{p} \frac{G_q^{(k)} L_{p-q}(u, v)}{(p - q)! q!} \right) \zeta^p \left( \sum_{j=0}^{\infty} w^j \frac{\zeta^{2j}}{j!} \right)$$

replacing $p$ with $p - 2j$ and equating the coefficients of $\zeta^p$, we obtain Equation (40).  □

**Theorem 10.** *For $u, v, w \in \Re$ and $p \geq 0$. Then,*

$$_L G_p^{(k)}(u, v + 1, w) = \sum_{q,j=0}^{p} \frac{p! (-u)^j {}_H G_{p-q-j}^{(k)}(v, w)}{(p - q - j)! q! (j!)^2} \tag{41}$$

**Proof.** By the definition of generalized Laguerre poly-Genocchi polynomials, we have

$$\sum_{p=0}^{\infty} {}_L G_p^{(k)}(u, v + 1, w) \frac{\zeta^p}{p!} = \frac{2 Li_k(1 - e^{-\zeta})}{e^\zeta + 1} e^{(v+1)\zeta + w\zeta^2} C_0(u\zeta)$$

$$= \left\{ \sum_{p=0}^{\infty} \left( \sum_{q=0}^{p} \frac{{}_H G_{p-q}^{(k)}(v, w)}{(p - q)! p!} \right) \zeta^p \right\} \left\{ \sum_{j=0}^{\infty} \frac{(-1)^j (u\zeta)^j}{(j!)^2} \right\}$$

$$= \left\{ \sum_{p=0}^{\infty} \left( \sum_{j=0}^{\infty} \sum_{q=0}^{p} \frac{(-u)^j {}_H G_{p-q}^{(k)}(v, w)}{(p - q)! p! (j!)^2} \right) \zeta^{p+j} \right\}$$

replacing $p$ with $p - j$, we have

$$\sum_{p=0}^{\infty} {}_L G_p^{(k)}(u, v + 1, w) \frac{\zeta^p}{p!} = \left\{ \sum_{p=0}^{\infty} \left( \sum_{q,j=0}^{p} \frac{(-u)^j {}_H G_{p-q}^{(k)}(v, w)}{(p - q)! p! (j!)^2} \right) \zeta^{p+j} \right\}$$

on equating the coefficients of $\zeta^p$, we obtain Equation (41).  □

**Theorem 11.** *For $u, v, w \in \Re$ and $p \geq 0$. Then,*

$$_LG_p^{(k)}(u, v+1, w) = \sum_{q=0}^{n} \binom{p}{q} {}_LG_{p-q}^{(k)}(u, v, w)$$

(42)

**Proof.** By the definition of generalized Laguerre poly-Genocchi polynomials, we have

$$\sum_{p=0}^{\infty} {}_LG_p^{(k)}(u, v+1, w)\frac{\zeta^p}{p!} - \sum_{p=0}^{\infty} {}_LG_p^{(k)}(u, v, w)\frac{\zeta^p}{p!}$$

$$= \left\{ \left( \frac{2Li_k(1 - e^{-\zeta})}{e^\zeta + 1} \right) e^{v\zeta + w\zeta^2} C_0(u\zeta) \right\} (e^\zeta - 1)$$

$$= \sum_{p=0}^{\infty} {}_LG_p^{(k)}(u, v, w)\frac{\zeta^p}{p!} \left( \sum_{q=0}^{\infty} \frac{\zeta^q}{q!} - 1 \right)$$

$$= \sum_{p=0}^{\infty} {}_LG_p^{(k)}(u, v, w)\frac{\zeta^p}{p!} \sum_{q=0}^{\infty} \frac{\zeta^q}{q!} - \sum_{p=0}^{\infty} {}_LG_p^{(k)}(u, v, w)\frac{\zeta^p}{p!}$$

$$= \sum_{p=0}^{\infty} \sum_{q=0}^{p} {}_LG_{p-q}^{(k)}(u, v, w)\frac{\zeta^p}{q!(p-q)!} - \sum_{p=0}^{\infty} {}_LG_p^{(k)}(u, v, w)\frac{\zeta^p}{p!}$$

Finally, equating the coefficients of the like powers of $\zeta^p$, we obtain the result of (42).  □

## 4. Addition-Symmetry Identities

Symmetry identities involving various polynomials have been presented (e.g., References [10,14–18,20–22]). As in the above-cited works, here we establish some addition-symmetry identities involving generalized Laguerre poly-Genocchi polynomials $_LG_p^{(k)}(u, v, w)$ in Equation (20).

**Theorem 12.** *Let $\alpha, \beta > 0$ and $\alpha \neq \beta$. For $u, v, w \in \Re$ and $p \geq 0$. Then,*

$$\sum_{q=0}^{p} \binom{p}{q} \beta^q \alpha^{p-q} {}_LG_{p-q}^{(k)}(u, \beta v, \beta^2 w) {}_LG_q^{(k)}(u, \alpha v, \alpha^2 w)$$

$$= \sum_{q=0}^{p} \binom{p}{q} \alpha^q \beta^{p-q} {}_LG_{p-q}^{(k)}(u, \alpha v, \alpha^2 w) {}_LG_q^{(k)}(u, \beta y, \beta^2 w)$$

(43)

**Proof.** Consider a function $h(\zeta)$, defined by

$$h(\zeta) = \left\{ \frac{(2Li_k(1 - e^{-\zeta})C_0(u\zeta))^2}{(e^{\alpha\zeta} + 1)(e^{\beta\zeta} + 1)} \right\} e^{\alpha\beta v\zeta + \alpha^2\beta^2 w\zeta^2}$$

(44)

we see that $h(\zeta)$ is symmetric with respect to the parameters or variables in each of the following pairs

$$(\alpha, \beta), \quad (u, v, w)$$

(45)

to produce the generalized Laguerre poly-Genochhi polynomials in Equation (20), we need to choose one element from each of the four pairs in Equation (45). There are 16 combinations consisting of those chosen four elements. We also need two combinations for function $h(\zeta)$. Therefore, we have two different cases for function $h(\zeta)$. For example, we have

$$h(\zeta) = \frac{1}{\alpha\beta} \sum_{p=0}^{\infty} {}_L G_p^{(k)}(u, \beta v, \beta^2 w) \frac{(\alpha\zeta)^p}{p!} \sum_{q=0}^{\infty} {}_L G_q^{(k)}(u, \alpha v, \alpha^2 w) \frac{(\beta\zeta)^q}{q!}$$

$$= \frac{1}{\alpha\beta} \sum_{p=0}^{\infty} \sum_{q=0}^{n} \binom{p}{q} \alpha^{p-q} \beta^q {}_L G_q^{(k)}(u, \beta v, \beta^2 w) {}_L G_{p-q}^{(k)}(u, \alpha v, \alpha^2 w) \zeta^p$$

similarly, we obtain

$$h(\zeta) = \frac{1}{\alpha\beta} \sum_{p=0}^{\infty} {}_L G_p^{(k)}(u, \alpha v, \alpha^2 w) \frac{(\beta\zeta)^p}{p!} \sum_{q=0}^{\infty} {}_L G_q^{(k)}(u, \beta v, \beta^2 z) \frac{(\alpha\zeta)^q}{q!}$$

$$= \frac{1}{\alpha\beta} \sum_{p=0}^{\infty} \sum_{q=0}^{n} \binom{p}{q} \alpha^q \beta^{p-q} {}_L G_{p-q}^{(k)}(u, \alpha v, \alpha^2 w) {}_L G_q^{(k)}(u, \beta v, \beta^2 w) \zeta^p$$

equating the coefficients of $\zeta^p$ on the right-hand sides of the last two equations we obtain desired Identity (43). $\square$

**Remark 1.** *On putting $\beta = 1$ in the above theorem, we obtain the following result:*

$$\sum_{q=0}^{p} \binom{p}{q} \alpha^{p-q} {}_L G_{p-q}^{(k)}(u, v, w) {}_L G_q^{(k)}(u, \alpha v, \alpha^2 w)$$

$$= \sum_{q=0}^{p} \binom{p}{q} \alpha^q {}_L G_{p-q}^{(k)}(u, \alpha v, \alpha^2 w) {}_L G_q^{(k)}(u, v, w) \tag{46}$$

**Theorem 13.** *Let $\alpha, \beta > 0$ and $\alpha \neq \beta$. For $u, v, w \in \Re$ and $p \geq 0$. Then,*

$$\sum_{q=0}^{p} \binom{p}{q} \sum_{i=0}^{\alpha-1} \sum_{j=0}^{\beta-1} \beta^q \alpha^{p-q} {}_L G_{p-q}^{(k)}\left(u, \beta v + \frac{\beta}{\alpha} i + j, \beta^2 x\right) {}_L G_q^{(k)}(u, \alpha w, \alpha^2 y)$$

$$= \sum_{q=0}^{p} \binom{p}{q} \sum_{i=0}^{\beta-1} \sum_{j=0}^{\alpha-1} \alpha^q \beta^{p-q} {}_L G_{p-q}^{(k)}\left(u, \alpha v + \frac{\alpha}{\beta} i + j, \alpha^2 x\right) {}_L G_q^{(k)}(u, \beta w, \beta^2 y) \tag{47}$$

**Proof.** Let

$$\begin{aligned} h(\zeta) &= \left\{ \frac{(2Li_k(1-e^{-\zeta})C_0(u\zeta))^2}{(e^{\alpha\zeta}+1)(e^{\beta\zeta}+1)} \right\} \left\{ \frac{(e^{\alpha\beta\zeta}-1)^2 e^{\alpha\beta(v+w)\zeta+\alpha^2\beta^2(x+y)\zeta^2}}{(e^{\alpha\zeta}-1)(e^{\beta\zeta}-1)} \right\} \\ &= \left\{ \frac{2Li_k(1-e^{-\zeta})C_0(u\zeta)}{(e^{\alpha\zeta}+1)} \right\} e^{\alpha\beta v\zeta+\alpha^2\beta^2 x\zeta^2} \left( \frac{e^{\alpha\beta\zeta}-1}{e^{\beta\zeta}-1} \right) \\ &\quad\times \left\{ \frac{2Li_k(1-e^{-\zeta})C_0(u\zeta)}{(e^{\beta\zeta}+1)} \right\} e^{\alpha\beta w\zeta+\alpha^2\beta^2 y\zeta^2} \left( \frac{e^{\alpha\beta\zeta}-1}{e^{\alpha\zeta}-1} \right) \\ &= \left\{ \frac{2Li_k(1-e^{-\zeta})C_0(u\zeta)}{(e^{\alpha\zeta}+1)} \right\} e^{\alpha\beta v\zeta+\alpha^2\beta^2 x\zeta^2} \sum_{i=0}^{\alpha-1} e^{\beta\zeta i} \\ &\quad\times \left\{ \frac{2Li_k(1-e^{-\zeta})C_0(u\zeta)}{(e^{\beta\zeta}+1)} \right\} e^{\alpha\beta w\zeta+\alpha^2\beta^2 y\zeta^2} \sum_{j=0}^{\beta-1} e^{\alpha\zeta j} \end{aligned} \tag{48}$$

$$= \left\{ \frac{2Li_k(1-e^{-\zeta})C_0(u\zeta)}{(e^{\alpha\zeta}+1)} \right\} e^{\alpha^2\beta^2x\zeta^2} \sum_{i=0}^{\alpha-1} \sum_{j=0}^{\beta-1} e^{(\beta v + \frac{\beta}{\alpha}i + j)\alpha\zeta} \sum_{q=0}^{\infty} {}_LG_q^{(k)}(u,\alpha w,\alpha^2 y) \frac{(\beta\zeta)^q}{q!}$$

$$= \frac{1}{\alpha\beta} \sum_{p=0}^{\infty} \sum_{i=0}^{\alpha-1} \sum_{j=0}^{\beta-1} {}_LG_{p-q}^{(k)}\left(u,\beta v + \frac{\beta}{\alpha}i + j, \beta^2 x\right) \frac{(\alpha\zeta)^p}{p!} \sum_{q=0}^{\infty} {}_LG_q^{(k)}(u,\alpha w,\alpha^2 y) \frac{(\beta\zeta)^q}{q!} \quad (49)$$

$$= \frac{1}{\alpha\beta} \sum_{p=0}^{\infty} \sum_{q=0}^{p} \begin{pmatrix} p \\ q \end{pmatrix} \sum_{i=0}^{\alpha-1} \sum_{j=0}^{\beta-1} {}_LG_{p-q}^{(k)}\left(u,\beta v + \frac{\beta}{\alpha}i + j, \beta^2 x\right) \sum_{q=0}^{\infty} {}_LG_q^{(k)}(u,\alpha w,\alpha^2 y)\beta^q\alpha^{p-q}\zeta^p$$

On the other hand,

$$h(\zeta) = \frac{1}{\alpha\beta} \sum_{p=0}^{\infty} \sum_{q=0}^{p} \begin{pmatrix} p \\ q \end{pmatrix} \sum_{i=0}^{\beta-1} \sum_{j=0}^{\alpha-1} {}_LG_{p-q}^{(k)}\left(u,\alpha v + \frac{\alpha}{\beta}i + j, \alpha^2 x\right) \sum_{q=0}^{\infty} {}_LG_q^{(k)}(u,\beta w,\beta^2 y)\beta^{p-q}\alpha^q\zeta^q \quad (50)$$

equating the coefficients of $\zeta^p$ in both sides of Equations (49) and (50), we obtain desired Identity (47). $\square$

**Theorem 14.** *Let $\alpha, \beta > 0$ and $\alpha \neq \beta$. For $u, v, w \in \Re$ and $p \geq 0$. Then,*

$$\sum_{q=0}^{p} \begin{pmatrix} p \\ q \end{pmatrix} \sum_{i=0}^{\alpha-1} \sum_{j=0}^{\beta-1} \beta^q\alpha^{p-q} {}_LG_{p-q}^{(k)}\left(u,\beta v + \frac{\beta}{\alpha}i + j, \beta^2 x\right) {}_LG_q^{(k)}\left(u,\alpha w + \frac{\alpha}{\beta}j, \alpha^2 y\right)$$

$$= \sum_{q=0}^{p} \begin{pmatrix} p \\ q \end{pmatrix} \sum_{i=0}^{\beta-1} \sum_{j=0}^{\alpha-1} \alpha^q\beta^{p-q} {}_LG_{p-q}^{(k)}\left(u,\alpha v + \frac{\alpha}{\beta}i + j, \alpha^2 x\right) {}_LG_q^{(k)}\left(u,\beta w + \frac{\beta}{\alpha}j, \beta^2 y\right) \quad (51)$$

**Proof.** The proof is similar to Theorem (4.2). Now, Equation (48) can also be expressed in the following form:

$$h(\zeta) = \frac{1}{\alpha\beta} \sum_{p=0}^{\infty} \sum_{i=0}^{\alpha-1} \sum_{j=0}^{\beta-1} {}_LG_{p-q}^{(k)}\left(u,\beta v + \frac{\beta}{\alpha}i + j, \beta^2 x\right) \frac{(\alpha\zeta)^p}{p!} \sum_{q=0}^{\infty} {}_LG_q^{(k)}\left(u,\alpha w + \frac{\alpha}{\beta}j, \alpha^2 y\right) \frac{(\beta\zeta)^q}{q!} \quad (52)$$

On the other hand, Equation (48) can be shown as equal to

$$h(\zeta) = \frac{1}{\alpha\beta} \sum_{p=0}^{\infty} \sum_{i=0}^{\beta-1} \sum_{j=0}^{\alpha-1} {}_LG_{p-q}^{(k)}\left(u,\alpha v + \frac{\alpha}{\beta}i + j, \alpha^2 x\right) \frac{(\beta\zeta)^p}{p!} \sum_{q=0}^{\infty} {}_LG_q^{(k)}\left(u,\beta w + \frac{\beta}{\alpha}j, \alpha^2 y\right) \frac{(\alpha\zeta)^q}{q!} \quad (53)$$

equating the coefficients of $\zeta^p$ to zero in the last two equations, we obtain desired Identity (51). $\square$

## 5. Connection with Sheffer Polynomials

One of the important classes of polynomial sequences is the class of Appell polynomial sequences. Appell polynomial sequences arise in numerous problems of applied mathematics, theoretical physics, approximation theory, and several other mathematical branches. The set of all Appell sequences is closed under the operation of umbral compositions of polynomial sequences. Under this operation, the set of all Appell sequences forms an abelian group. The Appell polynomial sequences are defined by the following generating function:

$$A(u,\zeta) = A(\zeta)e^{u\zeta} = \sum_{n=0}^{\infty} A_n(u)\frac{\zeta^n}{n!}. \quad (54)$$

power series $A(\zeta)$ is given by

$$A(\zeta) = A_0 + \frac{\zeta}{1!}A_1 + \frac{\zeta^2}{2!}A_2 + \cdots + \frac{\zeta^n}{n!}A_n + \cdots = \sum_{n=0}^{\infty} A_n\frac{\zeta^n}{n!}, \quad A_0 \neq 0, \quad (55)$$

where $A_i(i = 0, 1, 2, ...)$ are real coefficients. It is easy to see that, for any $A(\zeta)$, the derivative of $A_n(u)$ satisfies

$$A'_n(u) = nA_{n-1}(u) \tag{56}$$

Based on an appropriate selection for function $A(\zeta)$, different members belonging to the family of Appell polynomials can be obtained.

Khan, et al. [23] constructed a hybrid family, namely, Laguerre-Appell polynomials (LAPs) $_L A_n(u, v)$, as the discrete Laguerre convolution of the Appell polynomials. A systematic study of these polynomials is presented in Reference [24]. We recall that LAPs $_L A_n(u, v)$ are defined by means of the following generating equation:

$$A(\zeta)e^{v\zeta}C_0(u\zeta) = \sum_{n=0}^{\infty} {}_L A_n(u, v)\frac{\zeta^n}{n!} \tag{57}$$

we define the connection of generalized Laguerre poly-Genocchi polynomials $_L G_p^{(k)}(u, v, w)$ with Appell polynomials by means of the following generating function:

$$A(\zeta)e^{v\zeta + w\zeta^2}C_0(u\zeta) = \sum_{n=0}^{\infty} {}_L A_n(u, v, w)\frac{\zeta^n}{n!} \tag{58}$$

where

$$A(\zeta) = \frac{2Li_k(1 - e^{-\zeta})}{e^\zeta + 1}. \tag{59}$$

## 6. Concluding Remarks

The paper aimed at presenting the study of Laguerre poly-Genocchi polynomials, which play an important role in several fields of physics, applied mathematics, and engineering. These special polynomials are important as they possess essential properties such as recurrence and explicit relations, and functional and differential equations, summation formulae, and symmetric and convolution identities. These polynomials are useful and have potential for applications in numerous problems of number theory, combinatorics, classical and numerical analysis, theoretical physics, approximation theory, and other fields of pure and applied mathematics. The technique used here could further establish a wide variety of formulas for certain other special polynomials, and can be extended to derive new relations for conventional and generalized polynomial.

Furthermore, the generalized Laguerre poly-Genocchi polynomials $_L G_p^{(k)}(u, v, w)$ in Equation (20), being very general, can be specialized to yield various known polynomials and numbers, for example, Genocchi numbers $G_n$, Genocchi polynomials $G_n(x)$, and Hermite–Genocchi polynomials $_H G_n(u, v)$ (see Reference [12]). In this regard, the results presented here can be specialized to yield or be closely connected with some known identities and formulas (see, e.g., References [14–21,23,25]), and the references cited therein). Therefore, the results presented in this article could potentially be useful in problems that arise in the aforementioned fields.

**Author Contributions:** All authors contributed equally to this manuscript.

**Funding:** This research received no external funding.

**Acknowledgments:** The authors express their deep gratitude to the anonymous referees for their critical comments and suggestions to improve this paper to its current form.

**Conflicts of Interest:** The authors declare no conflict of interest.

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
