# Peer review of "A Study of Generalized Laguerre Poly-Genocchi Polynomials"

_mathematics, doi:10.3390/math7030219_

Round 1
Reviewer 1 Report
In Abstract and in the second item from the bottom of page 3 it is claimed that the research presented in the manuscript was motivated by "potential for applications ...". Me and also many of potential readers of the manuscript would like to see at least one specific potential application of the polynomials invented and studied in the manuscript. The authors over last ten years have published a bunch of papers (references [12]-[21] in the bibliography to the manuscript) by applying the same ideas to slightly different types of polynomials. Please give at least one reference to application of polynomials you have introduced in your published papers.
Author Response
Responses regarding Reviewer #1:
• The reviewer wants the conclusion point should be added to the manuscript. In this context, we incorporate the conclusion in the revised version of the manuscript.
• The reviewer wants at least one reference to the application of polynomials that I have introduced in my published paper. In this context, I have mentioned the following reference in the revised version of the manuscript, which is published in the Turkish Journal of Mathematics as follows:
Khan, N.U, Usman, T. and Choi, J: A new class of generalized polynomials associated with Laguerre and Bernoulli polynomials, Turkish J. Math., 43, (2019), 486-497.

Reviewer 2 Report
The paper provides a study of polynomials associated with the Genocchi family, the relevant properties have been derived and in the abstract there is a claim on the relevant usefulness in applications.
No mention of applications is then mentioned in the paper. This referee would like to know what applications (if any) the authors have in mind. My personal opinion is that although correct from the mathematical point of view it is difficult to find any relevant application.
Regarding the mathematical aspects this referee believes that the topic the Authors have discussed should be framed within the theory of Sheffer polynomials, which provides the natural context.
Regarding the references, the book by Appèl and Kampè dè Fèrièt should also be quoted along with 3 and 4 when regarding the two variable Hermite polynomials.
Furthermore the paper Sheffer polynomials, monomiality principle, algebraic methods and the theory of classical polynomials
G. Dattoli, M. Migliorati and H. M. Srivastava, Math. Comp. Modelling. 45 (2007) 1033
should be quoted, to get a better theoretical framing
Author Response
Responses regarding Reviewer #2:
Reviewer 2 suggested adding two references in the bibliography. In this context, I incorporate the following two references for the better theoretical framing of the manuscript.
Appell, P and Kampe de Feriet, J: Fonctions Hypergeometriques et Hyperspheriques Polynomes d Hermite, Gauthier-Villars, Paris, 1926.
Dattoli, G, Migliorati, M and Srivastava, H.M: Sheffer polynomials, monomiality principle, algebraic methods and the theory of classical polynomials, Math. Comp.Modelling. 45 (2007) 1033-1041.

Round 2
Reviewer 2 Report
This Referee has the impression that the Authors did not take care of the criticisms contained in the report.
What has been asked is not a mere inclusion of references but to provide an adequate discussion about the usefulness in Applications and the link with the Sheffer polynomial theory.
As this Referee has stressed is that this paper is very weak in its present form.
It does not contain any perspective and as it stands is no more than an exercise.
I regret but I cannot recommend its publication.
Author Response
Reviewer # 2
Comments and Suggestions for Authors
This Referee has the impression that the Authors did not take care of the criticisms contained in the report.
What has been asked is not a mere inclusion of references but to provide an adequate discussion about the usefulness in Applications and the link with the Sheffer polynomial theory.
As this Referee has stressed is that this paper is very weak in its present form.
It does not contain any perspective and as it stands is no more than an exercise.
I regret but I cannot recommend its publication.
Response
Thank you very much for the reviewer’s valuable comments to improve the paper in the current form. The authors revised the paper in the light of the reviewer’s comments. In this line, we included the following section in the revised manuscript, which is highlighted in the pdf.
One of the important classes of polynomial sequences is the class of Appell polynomial sequences. The Appell polynomial sequences arise in numerous problems of applied mathematics, theoretical physics, approximation theory and several other mathematical branches. The set of all Appell sequences is closed under the operation of umbral compositions of polynomial sequences. Under this operation, the set of all Appell sequences forms an abelian group. The Appell polynomial sequences are defined by the following generating function:
$$A(u,\zeta)=A(\zeta)e^{u\zeta}=\sum_{n=0}^{\infty}A_{n}(u)\frac{\zeta^{n}}{n!}.\eqno(5.1)$$
The power series $A(\zeta)$ is given by
$$A(\zeta)=A_{0}+\frac{\zeta}{1!}A_{1}+\frac{\zeta^{2}}{2!}A_{2}+\cdots+\frac{\zeta^{n}}{n!}A_{n}+\cdots=\sum_{n=0}^{\infty}A_{n}\frac{\zeta^{n}}{n!}, ~~~~A_{0}\neq0,\eqno(5.2)$$
where $A_{i}(i = 0, 1, 2, . . .)$ are real coefficients. It is easy to see that for any $A(\zeta)$, the derivative of
$A_{n}(u)$ satisfies
$$A^{'}_{n}(u)=n A_{n-1}(u)\eqno(5.3)$$
Based on appropriate selection for the function $A(\zeta)$, different members belonging to the family of Appell polynomials can be obtained.
Khan, et al. [24] constructed a hybrid family, namely, the Laguerre-Appell polynomials (LAP) ${}_{L}A_{n}(u,v)$ as the discrete Laguerre convolution of the Appell polynomials. A systematic study of these polynomials is presented in [13]. We recall that the LAP ${}_{L}A_{n}(u,v)$ are defined by means of the following generating equation:
$$A(\zeta)e^{v\zeta}C_{0}(u\zeta)=\sum_{n=0}^{\infty}{}_{L}A_{n}(u,v)\frac{\zeta^{n}}{n!}\eqno(5.4)$$
We define the connection of generalized Laguerre poly-Genocchi polynomials ${}_{L}G_{p}^{(k)}(u,v,w)$ with Appell polynomials by means of the following generating function:
$$A(\zeta)e^{v\zeta+w\zeta^{2}}C_{0}(u\zeta)=\sum_{n=0}^{\infty}{}_{L}A_{n}(u,v,w)\frac{\zeta^{n}}{n!}\eqno(5.5)$$
where
$$A(\zeta)=\frac{2Li_{k}(1-e^{-\zeta})}{e^{\zeta}+1}.\eqno(5.6)$$\\

Round 3
Reviewer 2 Report
The paper can be published.
The acknowledgment to the referee should be eliminated